# Controlled-Release Nanosystems with a Dual Function of Targeted Therapy and Radiotherapy in Colorectal Cancer

**DOI:** 10.3390/pharmaceutics14051095

**Published:** 2022-05-20

**Authors:** Pedro Cruz-Nova, Alejandra Ancira-Cortez, Guillermina Ferro-Flores, Blanca Ocampo-García, Brenda Gibbens-Bandala

**Affiliations:** Departamento de Materiales Radiactivos, Instituto Nacional de Investigaciones Nucleares, Carretera México-Toluca S/N, Ocoyoacac 52750, Mexico; pedro.cruz.comecyt@inin.gob.mx (P.C.-N.); alejandra.a.servicios@inin.gob.mx (A.A.-C.); guillermina.ferro@inin.gob.mx (G.F.-F.)

**Keywords:** nanoparticle system, colorectal cancer, combined therapy

## Abstract

Nanoparticles are excellent platforms for several biomedical applications, including cancer treatment. They can incorporate different molecules to produce combinations of chemotherapeutic agents, radionuclides, and targeting molecules to improve the therapeutic strategies against cancer. These specific nanosystems are designed to have minimal side effects on healthy cells and better treatment efficacy against cancer cells when compared to chemotherapeutics, external irradiation, or targeted radiotherapy alone. In colorectal cancer, some metal and polymeric nanoparticle platforms have been used to potentialize external radiation therapy and targeted drug delivery. Polymeric nanoparticles, liposomes, albumin-based nanoparticles, etc., conjugated with PEG and/or HLA, can be excellent platforms to increase blood circulation time and decrease side effects, in addition to the combination of chemo/radiotherapy, which increases therapeutic efficacy. Additionally, radiolabeled nanoparticles have been conjugated to target specific tissues and are mainly used as agents for diagnosis, drug/gene delivery systems, or plasmonic photothermal therapy enhancers. This review aims to analyze how nanosystems are shaping combinatorial therapy and evaluate their status in the treatment of colorectal cancer.

## 1. Introduction

Colorectal cancer (CRC) is one of the most aggressive types of cancer, representing the third most common cancer worldwide. CRC affects the epithelial cells that line the colon’s lumen and is progressively classified depending on the depth of invasion of original cancer, the involvement of lymph node metastasis, and the existence of distant metastasis. Therefore, CRC treatment and the corresponding therapeutic approaches depend on the tumor features and the patient response [1,2,3,4,5].

Common primary and metastatic CRC treatments include laparoscopic surgery for primary disease, resection of metastatic disease (e.g., liver and lungs), radiation therapy, neoadjuvant and palliative chemotherapy for rectal cancer, and some forms of metastatic disease [6]. The cytostatic agent 5-fluorouracil (5-FU), a thymidylate synthase inhibitor, was the first chemotherapeutic agent with demonstrable activity against colorectal cancer. Subsequently, the combination of 5-FU with leucovorin became the standard therapy for metastatic colorectal cancer, with a median survival of 8–9 months. The favorable interaction of oxaliplatin (a DNA cross-linking agent) with a 5-FU infusion led to the current standard combination called FOLFOX [7]. Irinotecan, a topoisomerase I inhibitor, was also combined with 5-FU in infusion with folinic acid and called FOLFIRI. The FOLFOX and FOLFIRI effect was greater than 5-FU or leucovorin alone and had similar efficacy, with a median survival of 18–20 months [8].

The major challenge in chemotherapy and radiotherapy is the lack of specificity. Immunotherapy for metastatic CRC has included the vascular endothelial growth factor receptor (VEGFR2) and epithelial growth factor receptor (EGFR). Bevacizumab, aflibercept, ramucirumab, cetuximab, and panitumumab are examples of targeted immunotherapy approved for use in metastatic colorectal cancer [9,10]. Nevertheless, with immunotherapy, the five-year survival rate for metastatic colorectal cancer is only slightly higher than 12%. The increase in response rates is due to various therapeutic strategies, such as monoclonal antibodies combined with chemotherapy. However, one of the main obstacles to this approach is the appearance of drug resistance [11]. Different chemotherapeutics have been loaded into nanoparticles (NPs) to improve their effectiveness in tumor growth inhibition, reduce the growth of metastatic tumors, overcome drug resistance, and increase the tolerable dose [12]. Targeted nanosystems can be used to interact with over-expressed active biological molecules on the surface of tumor cells, allowing the effective release of antitumor agents [13]. In addition, the incorporation of radionuclides into nanoparticles results in the formation of versatile nanosystem platforms with numerous applications for cancer detection and treatment [14]. Combinatorial therapy is a promising strategy for the synergistic treatment of several kinds of cancer. Due to the different mechanisms of action exhibited by the same system, it is possible to reduce the dosage of each agent in order to reduce the relative toxicity produced by the individual drugs in a conventional treatment [15], either with a radiopharmaceutical or chemotherapeutic immunotherapy and targeted therapy. This review aims to analyze how nano-sized approaches are shaping combinatorial therapy and their status in the treatment of colorectal cancer.

## 2. Hallmarks of Nanoparticle Systems

Targeting strategies of radionanoparticles or nanosystems are mainly focused on the inhibition of proliferation, induction of apoptosis, limiting metastasis and angiogenesis, and regulating the immune function of cancer cells by using biologically active molecules (peptides, antibodies, small molecules, etc.), also known as *ligands*. These specific nanoparticles are designed to have minimal side effects in healthy cells and better efficiency against cancer cells in comparison with free chemotherapeutics, nanoparticles, and/or nanosystems. In the nanosystems, physicochemical properties of bare nanoparticles, such as morphology, size, charge, and chemical composition, among others, determine their unique properties (optic, magnetic, luminescent, reactivity, ratio, surface–volume ratio, surface plasmon effect (SPR), etc.), while the ligands contribute to their colloidal stability (via electrostatic and/or steric interactions) and function (active targeting by specific affinity for the cell surface in cancer cells), and therefore, to the final size, shape, properties, and behavior (Table 1) [16,17,18,19].

In general, the use of nanoparticles for the diagnosis and treatment of CRC modifies the pharmacokinetic properties and the stability of the drug transported in order to achieve a controlled release at the desired specific sites, with the consequent decrease in toxicity over healthy tissues [20].

Depending on the nature of the nanoparticles (metallic, metal oxide, polymeric, carbon-based nanoparticles, etc.), different functionalization methods can be implemented. Physical methods include adsorption, interaction charge, and hydrophobic or hydrophilic interactions. A recurrent chemical procedure for the functionalization is the carbodiimide coupling to produce covalent bonds. In general, nanosystems provide novel opportunities and strategies for the imaging, detection, and diagnosis of CRC. Figure 1 shows the general mechanism of functionalization (physical and chemical).

## 3. Metallic Nanoparticles

### 3.1. Gold Nanoparticles

Gold nanoparticles (AuNPs) have been used as drug carriers due to their surface plasmon resonance and optical and other tunable properties. AuNPs can be prepared in a wide range of core sizes (from 1 to 150 nm), making it easy to control their dispersion [32]. The presence of negative charges on the AuNPs surface makes them easily modifiable, allowing them to be functionalized by the addition of several biomolecules such as drugs, targeted ligands, and genes. In addition, the biocompatibility and non-toxic nature of AuNPs make them an excellent candidate for their use as a nanoparticle system for combination therapy [33].

When metallic nanoparticles are prepared, colloidal stability needs to be reached. This feature is achieved by minimizing its surface charge, thus reducing the electrostatic interactions. For this purpose, small molecules with high redox potentials and the ability to form stable complexes are used, where sodium citrate (Na_3_C_6_H_5_O_7_) is the most-employed compound used for the stabilization of AuNPs.

Sodium citrate has been used in the AuNPs synthesis both as a reduction agent (stabilizer) and as a size regulator in the nucleation and growing phase (a process known as surface passivation). During the reduction and synthesis of AuNPs, the sodium citrate is adsorbed onto the surface of the gold nanoparticles.

The aforementioned interaction is accomplished through the carboxylate from the citrate group and the Au (111) plane of the NPs in three different modes (i) monocarboxylate bridging, (ii) monocarboxylate monodentate, and (iii) dicarboxylate bridging. The formed binding interactions are strong enough (6.7 kcal mol^−1^) to maintain the colloidal stability but weaker than a covalent Au-O bond (3 eV ≈ 70 kcal mol^−1^) and thus allow displacement by molecules that possess greater affinity (generally, biologically active molecules) [34].

At this point, a second chemical reaction is required to carry out the functionalization process itself. The formed bonds are most probably of covalent character; consequently, the nanosystems are chemically more stable. The chosen molecules exhibit a common characteristic, which is the presence of the sulfhydryl group (-SH) at the terminal end. This functional group reacts with metallic gold forming the Au–S covalent bond, with a binding bond of 40–50 kcal mol^−1^. AuNPs functionalized with thiolated ligands usually have excellent colloidal stability, with values of zeta potential > |30 mV|. The interaction between thiol-Au occurs with the thiol radical (deprotonated sulfhydryl group). Otherwise, if the thiol group is in its protonated form, the interaction will take place through the lone pair electrons of the sulfur, forming a less-stable coordination-type bond (Figure 2) [18,34,35].

Another way to functionalize AuNPs is by using amphiphilic molecules, which simultaneously act as stabilizers and ligands (Figure 3). Here, the vectors are adsorbed onto the AuNP surface. Additionally, ligands can form a bilayer or small micelles around the metallic nanoparticles [36].

#### AuNPs for Combined Therapy

One of the most widely used nanoparticles in combinatorial therapy systems are gold nanoparticles. AuNPs are versatile vehicles, commonly used in the active and passive release of molecules targeting specific tissues [37]. Therefore, AuNP systems were developed to enhance the anti-carcinogenic properties of the 17-allylamino-17-demethoxygeldanamycin (17-AAG), in combination with irradiation (IR), after inhibition of thermal shock protein 90 (Hsp90) in colorectal cancer cells. Hsp90 is a chaperone with an essential role in the organization and maturation of proteins involved in the survival and progression of cancer. 17-AAG is a potent Hsp90 inhibitor, as it inhibits its role as a chaperone, resulting in the degradation of Hsp90-dependent proteins [38]. The results indicated that irradiation alongside gold nanoparticles (GNP) and 17-AAG exposure combinations induced greater cytotoxicity than the treatments alone (*p* < 0.05). 17-AAG (9.45 nM) and GNP (70 μM) act as radiosensitizers in combination with irradiation. In addition, the triple combination of GNP, irradiation, and 17-AAG resulted in a significant decrease in cell viability when compared to the AuNPs and 17-AAG combination. Indeed, the triple combination of therapies (GNPs (70 μM), in combination with 17-AAG (4.72 nM) and X-ray irradiation 2 Gy)) significantly increased caspase 3 levels, suggesting a significant increase in apoptotic cells [39].

Controlled-release nanosystems allow the therapeutic agents to be carried directly into the tumor tissue, minimizing cytotoxicity in healthy tissues. They are also applicable to radiation therapy by limiting the dose of radiation to the tumor as much as possible [40]. Therefore, some researchers have suggested encapsulation of AuNPs in liposomes to increase the local radiation dose and a platinum-based drug as a chemotherapeutic agent (cisplatin, oxaliplatin, carboplatin) for the treatment of CRC. Activated platinum species react with nucleophilic centers at the N7 positions of guanosine and adenosine residues. The two reactive sites at the platinum core allow cross-linking between two adjacent guanines. Furthermore, platinum can be coordinated with the guanine bases of several DNA strands to form cross-bonds between strands [41]. Moreover, AuNPs provide a high density of low-energy electrons, which, together with chemotherapy, significantly and locally increase DNA damage. LipoGold liposomes were used to efficiently carry and release the AuNPs and chemotherapy into CRC cells to increase the likelihood that the combination of treatments will only locally damage tumor tissue. HCT116 colorectal cancer cells were treated with different levels of carboplatin, AuNPs, liposomes, and oxaliplatin, with a dose of 2 Gy of X-ray. Simultaneous administration of AuNPs and chemotherapy by liposomes is more effective than the same amount of these compounds without the carrier. In addition, the nanoparticle system was tested in vivo on a murine model with a subcutaneous xenograft of HCT116 cells. Carboplatin (at a low concentration), combined with AuNPs in a liposomal formulation, has a synergistic effect with a radiation dose of 10 Gy [42]. In addition to enhancing external radiation therapy treatment, strategies have also been implemented with AuNPs to enhance internal radiation therapy. Therefore, some authors have explored the potential of poly (methacrylic acid)-functionalized hybrid AuNPs to improve the effectiveness of radiotherapy with ^131^I in DHD/K12/TRb colorectal cancer cells. CRC cells were transfected with mNIS cDNA to express the iodine/sodium symporter (NIS) and were subsequently exposed to activities ranging from 0 to 11 MBq, in the presence or absence of perchlorate (NIS inhibitor). As a result, the 50% lethal dose was reached with 0.66 MBq. In addition, a significant reduction in cell viability was observed when incubated with AuNPs at 250 µg/mL. Subsequently, to determine if the AuNPs could enhance radiosensitivity to ^131^I, AuNP-^131^I was evaluated at predefined AuNP concentrations, using 0.1 MBq of ^131^I. In fact, at a AuNP concentration of 25 µg/mL, cell viability was reduced from 82% to 48% with ^131^I. Indeed, the AuNPs are an excellent tool to be used as enhancers of both external and internal radiation therapy due to their ability to interact with incident radiation and increase the deposited dose [43].

### 3.2. Hafnium Oxide Nanoparticles

Hafnium oxide (HfO_2_) is a material used for scintilligraphy applications due to its high density and high atomic number. Hafnium oxide nanoparticles (HfO_2_ NPs) have been reported as radiosensitizers, used to increase the dose and efficacy of radiotherapy inside tumors without causing any damage to surrounding healthy tissues [44]. Some techniques such as hydrothermal, solvothermal, and nonhydrolytic synthesis and precipitation methods have reported the successful obtention of HfO_2_ NPs. Their very small size allows them to accumulate inside cancer cells when they are injected into the desired tissue or organ of the human body. Thus, their use to improve the radiation dose for cancer treatment has been reported [45].

Hafnium oxide nanocrystals, identified as NBTXR3 nanoparticles, have been designed for efficient uptake of cancer cells and interaction with different types of ionizing radiation. This nanoscale, high-density material, and interaction with ionizing radiation allow the radiotherapy’s physical mode of action (formation of reactive oxygen species and DNA breakdown) to directly affect cancer cells [46].

#### Hafnium Oxide Nanosystems for Combined Therapy

NBTXR3 was clinically tested in the treatment of patients with locally advanced soft tissue sarcoma. For colorectal cancer, the appropriate concentration to kill HCT116 colorectal cancer cells is reported to be 800 µM. NBTXR3, in combination with radiation therapy (Ir), significantly decreases cell viability compared to radiation therapy alone (2 Gy, 96.3% vs. 2 Gy + NBTXR3, 89.3%). In addition, combination therapy significantly increases the break-up of the double DNA strand, as measured by histone H2AX phosphorylation, and increases micronuclei formation. In addition, NBTXR3 + radiation therapy significantly increases genome-derived cytosolic DNA, mediated by activation of the cGAS (cyclic GMP-AMP synthase)-STING G (stimulator of interferon genes) pathway [47] 96 h after treatment. Therefore, this demonstrates that NBTXR3, activated by Ir, may have an essential role in enhancing the antitumor immune response [48]. Another objective of nanoparticle systems with targeted therapy is to decrease the dose of radiation therapy in order to reduce adverse effects such as intestinal dysfunction, fecal incontinence, and pelvic fractures. DNA damage repair inhibitors such as Talazoparib (T) have been used to reduce the radiation dose. Talazoparib, a poly (ADP-ribose) polymerase (PARP) inhibitor, is a molecular analog of ADP ribose and inhibits the interaction between the PARP enzyme and ADP ribose, as well as acts as a PARP trap that prevents DNA repair, transcription, and replication, and induces direct-damage double-strand DNA breaks during the S-phase by stagnant replication fork collapse [49]. Organic metal frameworks such as hafnium (Hf) and 1,4-decarboxylase are a class of radiation-dose enhancers. The nanoparticles are coated with Fucoidan (Fuco), a natural polysaccharide derived from brown algae, which exhibits an affinity for P-selectin. This cell adhesion molecule is overexpressed in radiation-stimulated endothelial cells. After irradiating (IR) colorectal cancer cells with 2 Gy, the uptake of the Fuco-coated Hf-BDC nanoparticle system is significantly increased when compared to Fuco-free and IR-free formulations. Indeed, TT- Hf-BDC-Fuco + IR resulted in the highest cytotoxicity compared to all other treatment groups, suggesting that targeting P-sel improves efficacy [50].

A promising approach is the NBTXR3, proposed by Zhang and colleagues [51]. This nanosystem enables the reduction of colorectal cancer cell viability via an increase in radiation dose. It induces an antitumor immune response activation and produces a systemic anticancer response known as the abscopal effect, which allows localized radiation to induce an antitumor response at sites that were not exposed to RT. NBTXR3 was evaluated in CT26.WT colorectal cancer cells. NBTXR3, combined with RT (2 Gy), promotes early apoptosis. Importantly, NBTXR3 + RT increases CD8+ and macrophages (CD68+) in colorectal xenografts, which could be a solution to overcome the current limitations of radiotherapy [51,52].

## 4. Polymeric Nanoparticles

Unlike other nanoparticles, polymeric nanoparticles are defined in terms of their size as those particles with a size between 10 and 1000 nm (0.01 µm^−1^ µm) in at least one of their dimensions. They are usually employed for controlled and sustained drug delivery, mainly of drugs classified by the Biopharmaceutics Classification as types 2 and 4.

The methods for the synthesis of polymeric nanoparticles are based on the top-down (emulsion–evaporation, emulsion–diffusion, solvent displacement, and salting-out) and bottom-up (emulsion or microemulsion–polymerization, interfacial polymerization, and precipitation–polymerization) approaches. The first method uses dispersions of preformed polymers, while the second is based on the polymerization of the monomers up to the formation of the polymeric nanoparticles [53].

The methods followed by the incorporation of the drugs into the nanoparticles include adsorption, dissolution, entrapment, encapsulation, or chemical binding of drug molecules onto the nanoparticle surface. The selection criteria for one of the above-mentioned methods depend on the expected release kinetics and its characteristics since these are closely related to both the selected approach and the polymers that constitute the polymeric nanoparticles.

Among the easiest ways to functionalize and stabilize polymeric nanoparticles is using polyethylene glycol (PEG), which has been well-adapted for biomedical applications. For example, in bioconjugation and drug release, PEG is directly conjugated with drugs or to the surface of nanomaterials with encapsulated drugs to increase in vivo stability and solubility to reduce the elimination rate, thus optimizing the efficacy of the drug [54].

PEG is a polyether that prevents nonspecific adsorption of plasma proteins and uptake by nontarget organs, reduces immunological reactions, and also increases circulation time of the nanosystems and accumulation in target organs. It is usually used to coat polymeric nanoparticles or to functionalize inorganic NPs. This process is commonly called PEGylation, and the simplest way to achieve it is by adding PEG during nanoparticle synthesis. The interaction between PEG and the polymeric nanoparticles is possible because of the flexibility of the polymer chains and their high hydrophilicity [18,55]. The mechanism of colloidal stability of PEGylated nanoparticles is attributed to both entropic conformation and enthalpic effects of hydration [56,57].

As in the case of sodium citrate with AuNPs, PEG is weakly bound to the NPs, allowing its interchange for molecules with greater affinity. The employed PEG molecule can also be modified using head groups (thiols, amines, silanes, or carboxylic acids), which act as the linkers in a post-functionalization reaction. In this case, PEG acts as a bifunctional molecule (cross-linker) while maintaining the stability of the nanosystems [18].

A usual method of post-functionalization with modified PEG chains is the use of *N*-hydroxy succinimide ester (NHS-ester) moieties when the terminal groups contain carboxyl groups (carbodiimide coupling). This results in the formation of stable covalent amide bonds, and the mechanism of action is due to a rapid cross-linking between modified PEG and NHS-ester [55,58].

### 4.1. Poly (Lactic Acid) Nanosystems

Poly (lactic acid) (PLA) has already demonstrated potential as a drug carrier, which is an aliphatic thermoplastic polyester and one of the most-advanced bio-based, biodegradable, and biocompatible industrial polymers. It is obtained from raw materials such as cornstarch, sugar cane, and other renewable biomass products and excesses [59]. In addition, copolymerization, particularly with other polymers such as PEG, can help enhance biocompatibility. Therefore, in conjunction with the monoclonal antibody Cetuximab, the PEG–PLA nanoparticles have been used to carry the 5-FU chemotherapeutic agent and ^131^I to perform a combination therapy for colorectal cancer. Cetuximab is a monoclonal antibody used to treat patients with epidermal growth factor receptor-expressing RAS wild-type metastatic colorectal cancer. EGFR is a transmembrane glycoprotein, and its ligands are crucial for regulating cell proliferation, survival, and differentiation [60]. The Cetuximab (Cet)–PEG–PLA system does not exhibit marked cytotoxicity at a concentration of 1000 µg/mL per 24 h in colorectal cancer cells. However, Cet–PEG–PLA with 5-FU has shown to be more toxic than free 5-FU (5 µg/mL), attributed to the increased uptake [10]. Metabolized 5-FU has a higher affinity for thymidylate synthase than the dUMP natural substrate, which can prevent dTMP formation. Subsequent inhibition of DNA synthesis, combined with a consecutive synthesis of RNA and proteins, results in an imbalance that opposes cell survival [61]. Additionally, there was greater toxicity with the Cet–PEG–PLA-^131^I system when compared to free ^131^I at the same radiation activity (3.7 and 7.4 MBq/mL). However, the radio-chemotherapy combination Cet–PEG–PLA–5FU-^131^I (5 FU 0.625 µg/mL and 1.85 MBq/mL of ^131^I) resulted in significantly higher cytotoxicity when compared to monotherapy. Consequently, the Cet–PEG–PLA–5FU-^131^I nanoparticle system showed prolonged blood circulation and effective tumor accumulation [62].

### 4.2. Poly (Lactic-co-Glycolic Acid) Nanosystems

Poly (lactic-co-glycolic acid) (PLGA) is one of the most successfully used polymers in the biomedical field because it is easily biodegraded (by hydrolysis) into lactic acid and glycolic acid. These biocompatible monomers are metabolized by the tricarboxylic acid cycle for final excretion in the lungs [53,63].

The specific characteristics of PLGA nanoparticles (size, nanoparticle yield, nanoparticle loading) depend on the synthesis method employed (bottom-up or top-down), as well as parameters such as polymer molecular mass, polymer and copolymer concentration, surfactant concentration, and solvent used.

Advances in nanotechnology have revamped medical imaging. A wide range of materials has been designed to be incorporated into nanoparticles in order to serve as contrast agents for different imaging technologies [64]. Nanoparticles are also used to facilitate advanced diagnosis and monitor the effectiveness of the therapy applied. Nanoparticles can incorporate different contrast agents (e.g., radioactive, superparamagnetic, fluorescent), targeted combinations, and biocompatible sheets [65]. Almeida and collaborators (2020) developed a system consisting of nanoparticles of poly (D, L-lactide-glycolide) lactide: glycolide (PLGA) PVA/5-FU), labeled with ^99m^Tc for the early detection of colorectal adenocarcinoma. A subcutaneous CaCo-2 cell xenographic tumor model was used to corroborate that the nano-radiopharmaceutical had the desired functionality. After retro-orbital injection of the ^99m^Tc-PLGA/PVA/5FU system, it was possible to observe a high accumulation/uptake in the xenographic tumor. In addition, it was possible to observe high liver uptake, which was confirmed by the biodistribution profile. The results showed the specificity of the nanoparticle system. However, the system presents limitations, such as its high retention in the reticuloendothelial system [66].

### 4.3. Poly-ε-Caprolactone Nanosystems

Controlled-release nanosystems and targeted therapy are also used to overcome resistance to treatment, an increasingly common phenomenon in colorectal cancer. Hybridized nanoparticles, such as Dimethyldioctadecylammonium bromide (DDAB)-methoxy poly (ethylene glycol) (mPEG)-Poly-ε-caprolactone (PCL), are used as biodegradable nanoparticles with controlled-release kinetics, due to PCL. Controlled release nanosystems have been developed, together with gene therapy (siRNAs) that may promote apoptosis in Regorafenib-resistant cells. Regorafenib is a small, membrane-bound, intracellular, multi-kinase molecule inhibitor implicated in normal cell functions and pathological processes, such as oncogenesis, tumor angiogenesis, and the preservation of the tumor microenvironment [67]. Regorafenib/DDAB–mPEG–PCL, with anti-integrin-β1, was developed to inhibit the expression of integrin-β1 (A protein involved in cancer progression). Fifty nM of siRNA significantly decreased the level of integrin-β1 mRNA compared to the control group. This study shows that the inhibition of integrin-β1 expression by siRNA and using DDAB–mPEG–PCL as a carrier can potentially increase cell death by apoptosis of Regorafenib-resistant cells [68]. Moreover, PLG-coated nanoparticles have also been used for the combination of diagnosis and therapy. For example, Wang and collaborators (2018) used a poly (ethylene glycol)-ε-poly (caprolactone) block copolymer (PEG–PCL) for 5-FU co-charging and a plasmid encoding for green fluorescent protein (pEGFP). The study used two colorectal cancer cell lines (Caco-2 and SW480). Nanoparticles loaded with 5-FU have a significant efficiency (*p* < 0.05) in the cytotoxicity of cancer cells compared to 5-FU alone. In addition, the nanoparticle system, loaded with 5-FU alone and dually loaded with a plasmid, showed a significant decrease in tumor growth when compared to free 5-FU (tumor decrease percentage 69.6%, 71.3%, and 22.5%, respectively) in a mice SW480 cell tumor xenograft. Interestingly, the maximum plasma concentration of 5-FU in nanoparticles or in combination with plasmid was significantly higher than 5-FU alone (47, 49, and 13 µg/mL, respectively) [69].

### 4.4. Nanosystems Based on Polysaccharides

CRLX101 is a nanoapproach containing cyclodextrin and PEG conjugated to camptothecin (a topoisomerase inhibitor I), which was designed to overcome the problems associated with poor plasma solubility and increase in drug concentration in tumor tissue. The antitumor activity of CRLX101, both in monotherapy and in combination with other drugs, has been characterized by the enhanced efficacy when compared to camptothecin and irinotecan [70]. Weiss and collaborators (2013) demonstrated the safety, pharmacokinetics, and efficacy of CRLX101 in patients with advanced solid malignancies who enrolled in an open, single-arm, dose-escalation study. Sixty-two patients received treatment biweekly, which was generally well-tolerated [71]. The most common adverse effects were neutropenia and fatigue. In all treated patients, evidence of systemic plasma exposure to both polymer-conjugated and non-conjugated CRLX101 was observed. Moreover, CRLX101 improves chemoradiation therapy for rectal cancer. CRLX101 has been tested together with from 1 to 6 Gy of X-ray doses in HT-29 and SW480 colorectal cancer cells to determine the direct impact of cytotoxicity and radiosensitivity. The sensitization cell rate (dose required to have 10% survival) was 2.3 and 1.3, respectively. In addition, the antitumor potential of CRLX101 was evaluated along with a 5-FU regimen, with three daily fractions of X-ray radiation therapy (5 Gy × 3) in an SW480 cell tumor murine model. The highest therapeutic efficacy was observed with the treatment 5-FU + CRLX101 + XRT versus CRLX101 + XRT, with a *p*-value = 0.001. Additionally, the same model showed that the treatment did not present severe systemic toxicity 14 days after treatment [72]. In this sense, the combination of chemoradiotherapy with the CRL101 nanoparticle system has been taken to clinical phases. Toxicity and the complete pathological response rate of CRLX101 with standard chemoradiotherapy in locally advanced colorectal cancer have been evaluated. This open phase Ib/II study was initially designed to determine the recommended phase II (RP2D) dose of CRLX101 combined with chemoradiation therapy given at weeks 1, 3, and 5. Twenty-nine out of thirty-two patients had a complete clinical and pathological response. At the maximum tolerated weekly dose of 15 mg/m^2^ of CRLX101, two out of six patients achieved a complete pathological response. Therefore, combining treatments with CRLX101 nanoparticles in combination with chemoradiotherapy is a feasible strategy with an excellent toxicity profile [73].

### 4.5. Dendrimers

Dendrimers have a different conformation than other polymeric nanoparticles. They consist of globular molecules made of branches called *generations*. On their outer surface, dendrimers can be structured to have different functional groups, such as COOH, –COONa, –NH_2_, or –OH. Dendrimers are characterized by their molecular weight, highly branched three-dimensional spherical structure, and the ability to create a monodispersed medium. One of the most-employed dendrimers in drug delivery systems is PAMAM (polyamidoamine). PAMAM dendrimers have three mechanisms for the encapsulation of drugs: empty spaces (molecular encapsulation), branching points (hydrogen bonds), and outer surface groups (charge-charge interactions) [74,75,76].

In the treatment of colorectal cancer, small molecules have been identified as inhibitors of different types of proteins that play an important role in signaling pathways that promote cancer progression. The small molecule C19 inhibits the viability and proliferation of colorectal cancer cells in vitro and in vivo, through decreased activation of KRAS and, therefore, inhibition of phosphorylation of downstream kinases such as AKT and ERK [77,78,79]. In this sense, C19 has also been used for administration in pancreatic cancer via a PAMAM nanoparticle system for the combined treatment of targeted chemoradiation therapy. The nanosystem, identified as ^177^Lu-DN(C19)-CXCR4L, encapsulated with an efficiency of 81%. Indeed, the combination of C19 and a ^177^Lu radiation dose of 3 Gy/Bq promoted a significant decrease in viability and proliferation and led to the apoptosis of the radio-resistant and KRAS-dependent MIA PaCa-2 cell line, which expresses CXCR4 receptors [80]. Thus, the ^177^Lu-DN(C19)-CXCR4L nanosystem demonstrated a synergistic effect of chemo/radiation therapy, which could be implemented in colorectal cancer with mutated KRAS.

## 5. Liposomes

The structure of a liposome is described as a vesicle consisting of lipid bilayers that mimic cellular membranes. However, this similarity to cell membranes (without surface functionalization or conventional formulations) allows them to be rapidly captured by the mononuclear phagocytic system [16,19]. Due to the chemical and structural composition of the liposomes, one of their main functions is to serve as drug-delivery systems.

The general liposome method of preparation involves four basic stages: (i) drying down lipids from organic solvents, (ii) dispersing the lipid in aqueous media, (iii) purifying the resultant liposome, and (iv) analyzing the final product. These steps can be performed by passive (mechanical dispersion, solvent dispersion, detergent removal method) or active loading techniques [81].

The surface modification of liposomes with PEG decreases its rapid clearance since PEG creates a hydrophilic steric barrier around the liposomes, limiting opsonization and, thus, the rapid recognition of the nanosystems [16].

### Liposomes for Combined Therapy

Another example of PEG-coated nanoparticles with a combined effect are liposomes, which have been used to encapsulate Vinorelbine [5-nor-anhydrovinblastine (VNB)]. VNP is an antimitotic agent used against cancer, and its primary mechanism of action is related to the inhibition of microtubule dynamics that leads to mitotic arrest and cell death. As a microtubule destabilizing agent, VNP stimulates the depolymerization of microtubules and destroys the mitotic spindle at high concentrations while blocking mitotic progression at lower concentrations [82]. VNP encapsulated in liposomes and covered with PEG was combined with Auger electron radiation from ^111^In to complete the system. The ^111^In-VNB-liposomes system was tested on a murine model of a subcutaneous xenograft HT-29 colorectal cancer cell transfected with luciferase. The mice were treated once a week for one month. The in vivo bioluminescence visualization trial revealed that treatment with 5 mg/kg of ^111^In-VNB-liposomes showed a significant decrease in tumor growth, demonstrating its safety and reliability. In addition, the study showed that the combination of a low-dose radionuclide and a controlled anticancer agent could provide diagnostic and therapeutic efficacy [83].

## 6. Other Nanoparticles

### 6.1. Albumin-Based Nanoparticles

Albumin has been regarded as a promising material for producing nanoparticles for bioimaging and drug delivery because of its useful characteristics, including biocompatibility, biodegradability, non-toxicity, good water solubility, and easy handling. The nanosystems based on bovine serum albumin (BSA) and human serum albumin (HAS) are good candidates for intravenous use [18]. Albumin-based NPs have been demonstrated to be an effective drug-delivery system, due to the inherent targeting modalities of albumin, through gp60- and SPARC-mediated receptor endocytosis. An example of its use is in the approved formulation of Abraxane, an antineoplastic nanosystem for Paclitaxel delivery. Furthermore, surface functionalization was exploited for active targeting due to albumin’s abundance of carboxyl and amino groups [84,85].

Albumin nanoparticles can be synthesized utilizing various approaches (desolvation, thermally induced aggregation, self-assembly, and albumin-bound technology), and they offer useful properties, such as passive and active targeting, controlled drug release, and bypassing cancer drug resistance mechanisms [85].

#### Albumin-Based Nanoparticles for Combined Therapy

Among the existing options to overcome resistance to radiation therapy, the controlled-release systems loaded with gene-therapy agents aim to inhibit the expression of proteins that play an essential role in cancer. For example, the radiosensitization effect of the human serum albumin (HSA) nanoparticle system, loaded with a plasmid to inhibit proteins involved in chemoresistance and tumor growth expression, has been studied. Albumin is the most abundant protein in human serum (35–50 g/L). HSA binds to several drugs and impacts their distribution, metabolism, and therapeutic effect. Albumin has a molecular weight of approximately 66.5 kDa, which prolongs its circulation time when compared to other small molecules [86]. The HSA nanoparticle system can also be loaded with the anti-survivin plasmid. Survivin is a small protein that belongs to the family of apoptosis-inhibiting proteins. It is overexpressed in tumors of differentiated adult tissues. Thus, this apoptotic inhibitor plays an essential role in promoting the survival of cancer cells and inhibiting cell death [87]. The combined therapy of HSA nanoparticles loaded with an anti-survivin plasmid with single doses of radiation (0–8 Gy) was developed for effective inhibition of SW480 colorectal cancer cells. In fact, the cells treated with radiation doses of 2 Gy, together with HSA-anti-survivin, showed a decrease in viability of more than 60% with regard to the irradiated cells without the nanoparticle system. This implies that combination therapy is required, with the aim of decreasing the toxicity of targeted therapy (gene therapy) and enhancing the radiation therapy [88]. There has been research carried out on nanoparticle systems functionalized with ligands that can bind to membrane proteins overexpressed in radio-resistant cancer cells. In this regard, Lee and collaborators (2020) reported the identification of low-density lipoprotein receptor-related protein-1 (LRP1) as a biomarker of radio-resistant colorectal cancer cells in a model of a patient-derived xenograft (PDX). The response to radiation therapy was classified in PDX colorectal cancer mice. A total dose of 10 Gy (2 Gy × 5) or 0 Gy (control) was administered locally to the tumor. The tumors were classified as radio-sensitive or radio-resistant according to the growth inhibition rate. Therefore, the controlled-release system used, in combination with the recognition peptide (B5) for LRP1 (LDL-Receptor-Related Protein 1), were nanoparticles based on HSA. These nanoparticles were used to encapsulate 5-fluorouracil (HSA–5Fu). Compared to other treatments, the B5–HS–5Fu system with IR (2Gy × 5) showed significant tumor growth inhibition in a radio-resistant PDX model. These results demonstrate that the B5–HSA–5Fu system enhances the efficacy of neoadjuvant therapy in radio-resistant colorectal cancer [89]. Another example of combined therapy with albumin-based nanoparticles in colorectal cancer is the ^131^I-antiEGFR–BSA–PCL system based on bovine serum nanoparticles (BSA). This system consists mainly of liposomes loaded with ^131^I and functionalized with the antibody of the epidermal growth factor receptor. This system improves efficacy and reduces radionuclide toxicity in diagnosis and treatment. It was tested on a murine model of LS180 colorectal cancer cells to determine whether the nanoparticle system effectively decreases radionuclide toxicity. Both the ^131^I-antiEGFR–BSA–PCL and ^131^I-BSA–PCL systems did not show toxicity, avoiding the mice’s body weight decreasing or promoting liver or kidney degradation when compared to the treatment with ^131^I alone or saline solution. In addition, the ^131^I-antiEGFR-BSA-PCL and ^131^I-BSA–PCL nanoparticle systems significantly decreased tumor growth when compared to the two other treatments. However, differences between nanoparticle systems were not significant in terms of tumor growth reduction. This nanoparticle system may be a good candidate for reducing the side effects of radiation therapy in colorectal cancer [90]. Because of its intrinsic properties, BSA can be used for nanoparticle systems in combination therapy. BSA and bismuth sulfide (Bi_2_S_3_) have also been used to provide stability and increase the half-life of nanoparticles in the bloodstream. Nanoparticles loaded with the chemotherapeutic methotrexate (DNA synthesis inhibitor) are excellent candidates for combined therapy with radiation therapy. The Bi_2_S_3_–BSA–methotrexate (MTX) nanoparticle system was tested on the SW480 colorectal cancer cell line with and without radiotherapy. The feasibility test showed no significant differences between the treatment with MTX alone or loaded within the Bi_2_S_3_–BSA nanoparticles. However, when cells were exposed to X-rays, only the cells treated with the nanoparticle system exhibited a significant reduction in cell viability. Therefore, the system is likely to possess protective properties against the side effects of MTX and enhance the effect of radiation therapy due to the property of Bi_2_S_3_ being a metal compound with a high atomic number. In addition, colorectal cancer cells treated with the Bi_2_S_3_–BSA–MTX system, combined with X-rays, showed a high percentage of cell death from apoptosis and a low percentage of cell death from necrosis [91].

### 6.2. Hyaluronic-Acid-Coated Nanoparticles

Hyaluronic acid (HA) is a natural mucopolysaccharide and is an excellent tool in pharmacology as it exhibits crucial characteristics such as biocompatibility and non-immunogenicity, has chemical versatility and non-toxicity, is biodegradable, and has a high hydrophobicity. In addition, several types of tumors overexpress receptors that have a high affinity for HA [92]. HA, composed of alternating D-glucuronic acid and *N*-acetyl-D-glucosamine, is a negatively charged polymer that acts as a recognition agent for the CD44 membrane receptor (overexpressed in colorectal cancer). In addition to overcoming the phenomenon of radio-resistance, the main objective of nanoparticle systems is to transport chemotherapeutics and extend their circulation time, which allows control over the kinetics of release and is a specific cellular targeting medium.

#### Hyaluronic Acid (HA)-Coated Nanoparticles for Combined Therapy

One of the strategies that have been used to achieve this purpose in colorectal cancer is to encapsulate the chemotherapeutic Raltitrexed (a thymidylate synthase inhibitor). The combination of hyaluronic acid and nanoparticles has been used, employing the layer-by-layer deposition technique. The HA–Raltitrexed nanoparticle system, combined with radiation therapy (4 Gy), showed statistically significant DNA damage (measured by γH2AX detection) when compared to individual treatments and control. The system was tested in vivo on a xenograft murine model. Mice were treated with the HA–Raltitrexed-IR system with three doses of 2 Gy. Treatment with HA–Raltitrexed–IR showed a significant decrease in tumor growth with regard to HA–Raltitrexed. However, treatment with IR alone showed similar results to the treatment with HA–Raltitrexed–IR. The authors conclude that after optimizing the dosage and time of IR and HA–Raltitrexed, this nanosystem combined with radiotherapy will potentially translate into clinical trials [93]. The high binding affinity of hyaluronic acid to CD44 receptors overexpressed on the surfaces of colorectal cancer cells made it a potential system for the administration of anticancer drugs. Lipid–polymer hybrid nanoparticles (LPNPs) with hyaluronic acid (HA) have also been used to encapsulate Irinotecan (I) together with a DNA plasmid (D) for gene therapy. In vitro cytotoxicity studies show that the HA-I-D-LPNPs system significantly decreases the viability of SW480 colorectal cancer cells when compared to gene therapy or Irinotecan alone. Non-loaded LPNs showed no cytotoxicity, indicating the safety and biocompatibility of the system. Tumor volume (subcutaneous xenograft with SW480 cells) of mice treated with the HA-I-D-LPNPs system was reported to be lower than those treated with I-D-LPNPs alone. The authors conclude that this can be explained by CD44 receptor-mediated targeting, indicating that the HA-I-D-LPNPs system can significantly inhibit tumor growth, which is consistent with the results of the cytotoxicity test [94].

Table 2 summarizes nanoparticle systems for chemo- and radiotherapy targeted toward colorectal cancer.

Figure 4 shows nanoparticle systems for combined chemoradiotherapy in colorectal cancer. Chemotherapeutical may be inside or outside the nanoparticle and carried to the cancer cell through ligands, antibodies, and/or molecular recognition agents that interact with overexpressed proteins in colorectal cancer tumors. The combination with radiation therapy can be through external X-rays and also by using radiolabeled molecules.

## 7. Radiolabeling Strategies

The aforementioned systems can also incorporate radionuclides to provide diagnostic and/or treatment tools that, together with conventional therapy, represent a new strategy of combined therapy based on a nanoparticle platform.

Among the objectives during the development of radiolabeled nanoparticles, some specific objectives are to maximize the binding affinity to a specific target and to develop complementary labeled imaging agents that also possess therapeutic properties.

Designing multimeric systems provide a specific ligand–receptor union. Thus, molecules such as peptides, antibodies, small molecules, or any other molecule with biological activity can be linked to the surface of metal nanoparticles, polymeric nanoparticles, and liposomes, among others, to improve imaging of tumors overexpressing specific antigens or receptors.

Moreover, it is possible that the engineering of radiolabeled nanoparticles provides dually labeled imaging agents that target the same ligand. This allows cross-validation between different imaging approaches such as fluorescence, optical and nuclear images, nuclear and MRI (magnetic resonance images), or trimodal nuclear-MRI-fluorescence images.

Additionally, radiolabeled nanoparticles can exhibit therapeutic properties, not only by the presence of alpha or beta emitters but by external stimuli or the presence of drug molecules that combine therapeutic properties.

This radiopharmaceutical agent would be capable of simultaneously functioning as both a targeted radiotherapy system and a photothermal therapy system.

### General Aspects of Radiolabeled Nanosystems

The physicochemical properties of nanosystems such as size, surface chemistry, charge, shape, and multivalency, among others, determine particle–cell interactions, pharmacokinetics, and optical properties. The design of nanoparticles can consider the development of multimeric systems to produce multivalent effects, which amplify the strength of the binding process and can regulate various cellular processes.

The synthesis of nanoparticles with a variety of physicochemical properties has led to important advances in biomedical applications. Some of the most studied nanoparticles are those consisting of gold and iron oxide, as well as other structures, such as single-wall carbon nanotubes, fullerenes, multi-wall carbon nanotubes, etc.

Gold nanoparticles are relevant in the use of PET/CT or SPECT/CT imaging due to the relative ease of surface modification and radiolabeling, as well as biocompatibility and optical properties, while radiolabeled iron oxide nanoparticles have been designed for use in PET/MRI and SPECT/MRI dual techniques.

Radiolabeled nanoparticles have been conjugated to target specific molecules and have mainly been employed as agents for diagnosis, as well as studies for biokinetics of new devices based on nanoparticles that comprise radiopharmaceuticals, drug/gene delivery systems, or plasmonic photothermal therapy enhancers. The most common radionuclides for PET imaging are gallium-68 (^68^Ga), fluorine-18 (^18^F), carbon-11 (^11^C), and copper-64 (^64^Cu), and the most common SPECT imaging radionuclides include technetium-99m (^99m^Tc) and indium-111 (^111^In) [95,96].

The general structure of radiolabeled nanoparticles presents the following main components: inner core, targeting biomolecule (BM) (with high affinity for target epitopes), pharmacokinetic-modifying (PKM) linker, bifunctional coupling or chelating agent (BFC), and a radiotracer group, Figure 5 [95,96].

Radionuclides can be conjugated directly onto the NP surface in the presence or no presence of a spacer (hydrocarbon chain, a peptide sequence or a poly-ethyleneglycol linker) or can be attached to the NP during chemical synthesis.

Several strategies have been applied to the preparation of radiolabeled nanosystems. Radiolabeling approaches can be divided into three categories: (i) surface coupling (coordination of radioactive elements to the outer surface of the nanomaterials: indirect surface labeling using a chelator, indirect surface labeling using a prosthetic group, and direct surface labeling), (ii) inner incorporation (incorporation of radioactive elements inside nanomaterials: radiochemical doping, encapsulation, non-radioactive variant activation, isotope exchange, and non-isotope exchange), and (iii) interface engineering (labeling of radionuclides at the interface between nanomaterials and surface ligands), such as polyethylene glycol (PEG)ylation. Nevertheless, to take advantage of these properties, there are some requirements to avoid recognition by the mononuclear phagocyte system (MPS), followed by relatively long blood circulation times [97].

Representative radiolabeled nanosystems for colorectal cancer are presented in Table 3.

## 8. Potential Clinical Applications

Several anticancer nanosystems have been translated to the clinic. Almost thirty years ago, the liposome/doxorubicin nanosystem was approved by the FDA for cancer treatment, which has demonstrated improved efficacy and safety over standard and single therapies [109]. Since then, at least ten controlled-release nanosystems based on polymeric nanoparticles, micelles, and liposomes have been approved for clinical application, while several more are in clinical phase II and phase III [110]. According to The ClinicalTrials.gov registry, several nanosystems for the delivery of chemotherapeutic drugs for colorectal cancer such as cetuximab loaded into targeted polymeric NP (Phase I), SNB-01 containing the active metabolite of irinotecan (Phase I), folate functionalized nanocarrier for a topoisomerase-1 inhibitor delivery (Phase 2), and the Irinotecan liposome formulation (phase 2) are under clinical evaluation. Likewise, the CRLX-101 for capecitabine delivery, when combined with conventional therapy and radiation external therapy, is under phase I evaluation.

In particular, nanosystems with a dual function of targeted therapy and radiotherapy in colorectal cancer are still under a preclinical stage of research. However, many of the drug delivery radionanosystems previously described in this review have the advantage of being natural or biodegradable vectors that may give the added benefit needed for clinical translation. In other cases, the background of successful radiotherapeutic results has allowed the first application in humans. For example, ^177^Lu-peptides for targeted radiotherapies of prostate cancer or neuroendocrine tumors [111,112] served as clinical experience to administer ^177^Lu_2_O_3_-peptide nanoparticles in a patient with colorectal liver metastases [107]. There is no doubt that the preclinical advances in nanosystems shown in this review make it clear that radionanosystems for targeted therapy and radiotherapy of colorectal cancer are on the right path to clinical translation.

## 9. Conclusions

The use of nanoparticles in colorectal cancer treatment has become an interesting approach to be used for producing more than one modality of therapy. They offer excellent platforms for tumor targeting and drug/gene delivery for selective treatments. Moreover, nanoparticles can be formulated to produce chemo or radiotherapy and to combine the therapeutic modes aimed to produce a synergistic effect and, in this way, improve the efficacy of a single treatment. Nanoparticles for combinatorial therapies represent a new tool for improving the treatment of colorectal cancer. Here, we have attempted to summarize the recent contributions to nano-sized targeted nanoparticles for drug or gene delivery for combinatorial therapies on colorectal cancer.

## Figures and Tables

**Figure 1 pharmaceutics-14-01095-f001:**
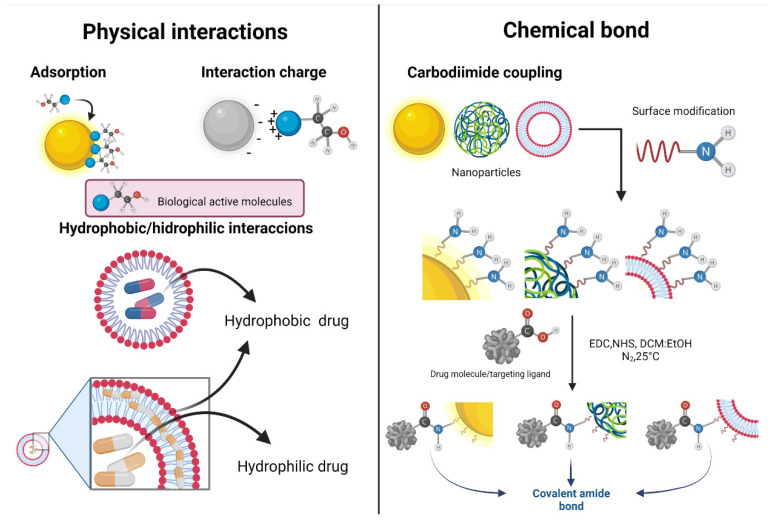
Physical and chemical interactions of nanoparticles for functionalization, drug loading, or radiolabeling. Physical methods include adsorption, interaction charge, and hydrophobic or hydrophilic interactions. Chemical bonds can be formed by carbodiimide coupling.

**Figure 2 pharmaceutics-14-01095-f002:**
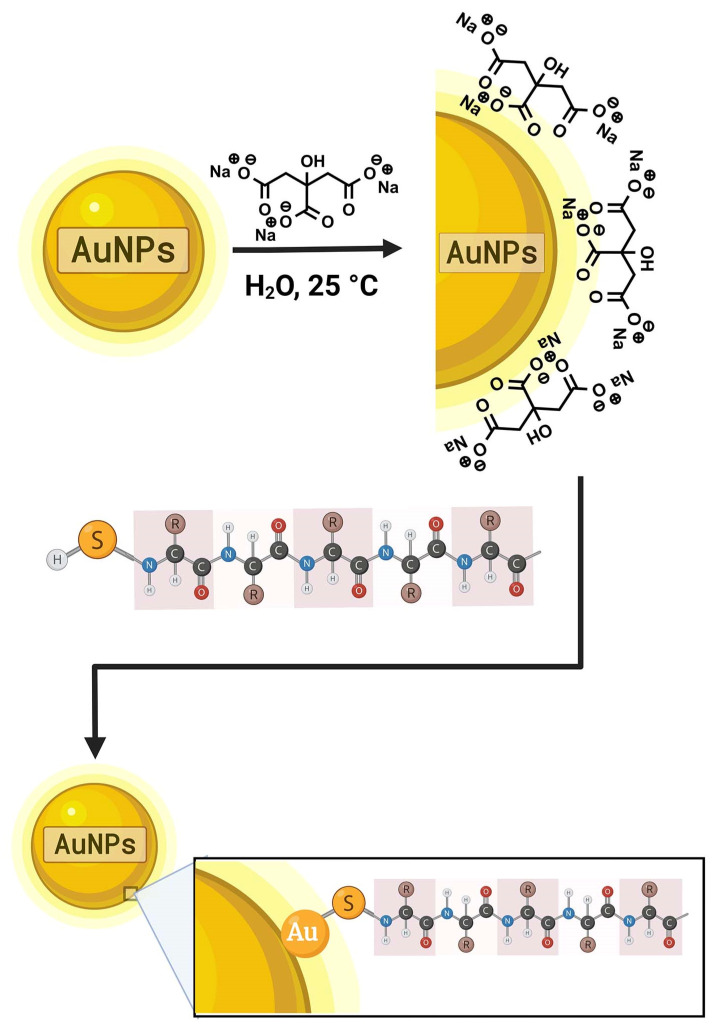
Mechanism of gold nanoparticles functionalization by covalent bonding. The sodium citrate acts as a stabilizer when it is absorbed into AuNPs. Subsequently, a second chemical reaction with the sulfhydryl group of functionalization molecules reacts with metallic gold forming the Au–S covalent bond.

**Figure 3 pharmaceutics-14-01095-f003:**
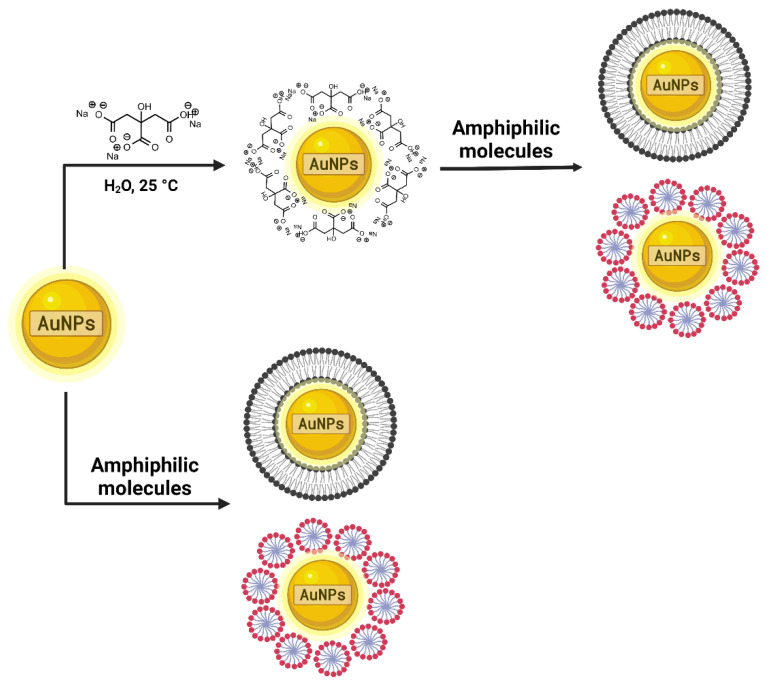
AuNP functionalization with amphiphilic molecules. Amphiphilic molecules (bilayer or small micelles) are absorbed into the AuNp surface and can act as stabilizers or ligands.

**Figure 4 pharmaceutics-14-01095-f004:**
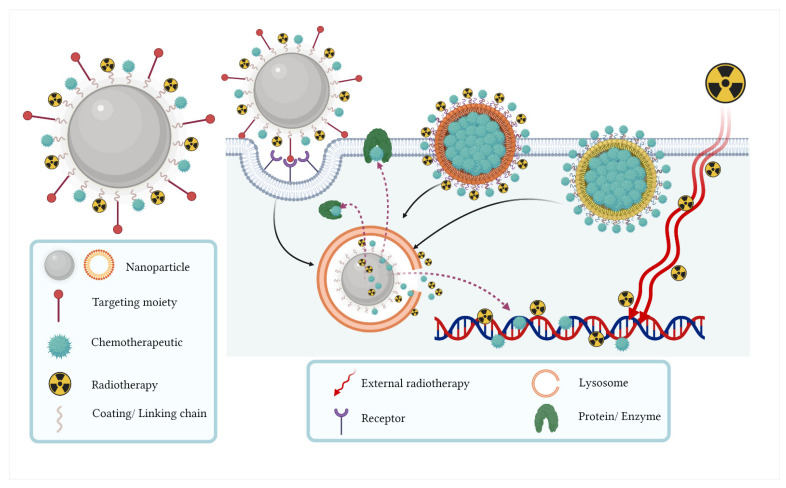
Nanoparticle systems for combined chemoradiotherapy in colorectal cancer. The chemotherapeutic agent can be loaded or conjugated to the nanoparticle and can be carried to the cancer cell through ligands, antibodies, and/or molecular recognition agents that interact with overexpressed proteins in colorectal cancer tumors. The combination with radiation therapy can be through external X-rays or by using radiolabeled molecules.

**Figure 5 pharmaceutics-14-01095-f005:**
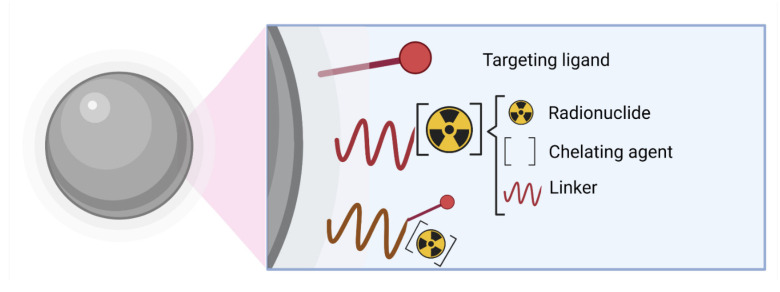
Main components of targeted radiolabeled nanoparticles. Radionuclides can be conjugated directly onto the nanoparticle surface, with or without a spacer.

**Table 1 pharmaceutics-14-01095-t001:** Importance of nanoparticle properties in biological behavior.

Characteristic/Property	Importance	Biological Effect	Reference
Particle size	Particle size determines and affects essential parameters of the nanoparticles, such as surface reactivity, and therefore determines their cellular uptake and clearance. Moreover, the enhanced permeability and retention (EPR) effect is another dependent-size property of nanoparticles.	NPs with sizes less than 100 nm have 2.5-fold greater uptake than those of a 1 µm diameter. Nanoparticles with sizes less than 10 nm have better clearance. Additionally, particles with larger dimensions have greater impact on cellular functions (cell proliferation, apoptosis, cytoskeleton formation, adhesion, and migration).	[18,20,21,22,23]
Shape	The nanoparticle uptake has been shown to be a phenomenon dependent on their shape (rods, start, nanotubes, triangle, spherical, etc.). Mainly in bare metallic nanoparticles, this feature determines their physicochemical properties.	Generally, spherical particles have shown better cellular uptake. The synthesis methods are also easier.	[24,25,26,27]
Surface–volume ratio	Surface–volume ratio affects the nanoparticle internalization or the drug internalization.	NPs with a greater surface–volume ratio have better internalization due to the improved possible interactions with the surface of the cells.	[20]
Surface charge	The surface charge influences solubility, stability, cellular uptake, and the cytotoxicity of nanoparticles.	NPs with positive charge have better internalization in vitro due to the characteristics of the cell membrane.	[20,24,27,28]
Chemical composition	Metallic	Metallic NPs exhibit photothermal, magnetic, optical, and luminescent properties, among others, useful in therapeutic and diagnostic approaches.	[29]
	Polymeric	Usually, polymeric NPs have been used for targeted delivery and controlled drug release.	[30]
	Liposomes	Some nanosystems based on liposomes have been approved by the FDA for the use as drug-delivery vehicles.	[31]

**Table 2 pharmaceutics-14-01095-t002:** Nanoparticles carriers of chemotherapeutic agents for colorectal cancer.

Nanoparticle	Chemotherapy or Targeted Therapy	Radiotherapy	Reference
AuNPs	17-AAG	2 Gy of X-ray IR	[39]
LipoGold	Carboplatin, oxaliplatin	2 and 10 Gy of X-ray IR	[42]
NBTXR3	Vía cGAS-STING	1–4 Gy as a single dose.	[48]
Hf-BDC	Talazoparib and Fucoidan	2 Gy as a single dose	[50]
PEG–PLA	Cetuximab and 5-FU	^131^I	[62]
HAS–PEG	Anti-survivin miRNA	2–8 Gy as a single dose	[88]
PEG–Liposomes	Vinorelbine	^111^In	[83]
HAS	B5 (Peptide to bind LRP1) and 5-FU	10 Gy as a single dose.	[89]
BSA–PCL	Anti-EGFR	^131^I	[90]
BSA–Bi_2_S_3_	Methotrexate	X-ray at a final dose of 2 Gy	[91]
HA	Raltitrexed and CD44 targeting	2 Gy as a single dose	[93]
CRLX101	Camptothecin and 5-FU	1–6 Gy of X-ray	[72]

**Table 3 pharmaceutics-14-01095-t003:** Radiolabeled nanosystems for colorectal cancer.

Radioisotope	Radiolabeled Approach	Imaging Application/Study	References
^64^Cu	Au nano shell/cyclic-RGD/DOTA	PET imaging and thermoablation treatment in HCT116 human colorectal cancer xenografted mice.	[98]
	^64^Cu-loaded PEGylated liposomes	MicroPET/CT imaging in implanted colon adenocarcinoma in mouse model.	[99]
^68^Ga	AuNPs/NOC, TOC with DOTA as chelator ^68^Ga-DOTA-TOC, ^68^GaDOTA-NOC	Evaluation of using gold radionanoplatforms to enhance the intracellular retention of ^68^Ga in tumor cells (CRC cell line, HT-29) with regard to the use of congener radiocomplexes carrying the somatostatin analogs Tyr(3)-octreotide (TOC) and Nal(3)-octreotide (NOC).	[100]
	PEG-gold nanoparticles with ^68^Ga-DOTA-Neuromedin B, ^68^Ga-DOTA-PEG(4)-BBN(7–14), ^68^Ga-DOTA-NT, and ^68^Ga-DOTA-Neuromedin N	Preclinical evaluation (in vitro binding kinetics and specific binding) of Gold NPs functionalized with bombesin or neurotensin-like peptides for the targeting of colon tumors (human HT-29 colon carcinoma).	[101]
^124^I	^124^I-Au@AuCBs	Assessment of crushed gold-shell radioactive nanoballs with theranostic potential in photothermal therapy, based on macrophage-mediated delivery of the NPs to the tumor tissues. The capability of ^124^I-Au@AuCBs to enhance photodynamic therapy in colon-cancer-bearing mice, when administered intratumorally, was demonstrated.	[102]
^131^I	Cetuximab [Cet]–PEG–PLA–5Fu-^131^I NPs	Assessment of combined therapeutic effects of ^131^I -labeled and 5Fu-loaded multifunctional NPS in CRC (spherical and pH-sensitive) on cell viability and apoptosis in CRC.	[62]
	(PMAA–AuNPs) combined with radioiodine	In vitro clonogenic assays performed on CRC cells showed that poly(methacrylic acid)-grafted gold nanoparticles (PMAA–AuNPs) could efficiently lead to marked tumor cell mortality when combined with a low activity of radioiodine, which alone appeared to be essentially ineffective on tumor cells.	[43]
^99m^Tc	^99m^Tc–radiolabeled 2c-NP.	A betulinic acid analog (2c) was encapsulated within a polymeric nanocarrier system (PLGA/PVA) (2c-NP) and evaluated for in vitro and in vivo therapeutic efficacy. ^99m^Tc was used to visualize in vivo localization of 2c-NP in CRC animal models.	[103]
^99m^Tc	^99m^Tc–Resveratrol–AuNP	^99m^Tc-labeled gold nanoparticles and surface coating of resveratrol, purposed for the non-invasive detection of colon cancer by SPECT imaging. Presence of resveratrol increases their selectivity towards CRC cells. The accumulation of ^99m^Tc–Res–AuNP in HT-29 cells was significantly higher than in the analog non-targeted system.	[104]
^99m^Tc	^99m^Tc–EuB–FA–SLN–OP	Development of Eudragit S100–coated, pH-awakened microbeads (MBs), encapsulating folic acid (FA)-modified tristearin solid lipid nanoparticles (SLNs), loaded with oxaliplatin (OP). Development of MB formulation (^99m^Tc–EuB–FA–SLN–OP) suggested promising results against therapy of CRC, using dual targeting (i.e., ligand-directed and pH-awakened) approach.	[105]
^177^Lu	^177^Lu–EuDPA/SiO_2_–NH_2_	^177^Lu-Labeled Eu-Doped mesoporous SiO_2_ NPs as a theranostic radiopharmaceutical for CRC. Radioactive EuDPA/SiO_2_-NH_2_ NPs were obtained by replacing Eu^3+^ sites with ^177^Lu, allowing the formation of a dual-modality probe for both SPECT imaging and clinical radiotherapy of CRC.	[106]
	^177^Lu_2_O_3_-iPSMA/-iFAP	Specific uptake in colorectal cancer metastasis. Nanosystem designed for targeted radiotherapy with dual modality for both SPECT imaging and radiotherapy in CRC.	[107]
^188^Re	^188^Re-BMEDA-labeled PEGylated liposomes	Development of *N N*-bis(2-mercaptoethyl)-*N′*,*N′*-diethylethylenediamine (BMEDA)-labeled PEGylated liposomes. In vivo evaluation in murine C26-colon tumor-bearing mice with MicroSPECT/CT imaging analysis revealed that liposomes had a higher tumor uptake when compared to unencapsulated ^188^Re-BMEDA. Posterior comparison between PEGylated liposomes, ^188^Re-liposomes and Doxcorubicin-loaded ^188^Re-doxorubicin-liposomes.	[108]
^111^In		In vivo therapeutic efficacy evaluation of dual-nanoliposome encapsulated vinorelbine (VNB) and ^111^In-oxine on HT-29/luc mouse xenografts.	[83]

## Data Availability

Not applicable.

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
