# Peer review of "Controlled-Release Nanosystems with a Dual Function of Targeted Therapy and Radiotherapy in Colorectal Cancer"

_pharmaceutics, 2022, doi:10.3390/pharmaceutics14051095_

Round 1
Reviewer 1 Report
Overall think this paper is fantastically composed. The writing is informative and the figures are beautifully presented with a good level of detail. I have a few points I would like to see addressed to make this paper the best it can be.
1) The is a lot of great references to pre-clinical work with cell lines and mice, but I, the reader, want to know more about the current clinical landscape of nanoparticles for radiation therapy. Are they being used clinically at the moment? If not, why not? Are there ongoing clinical trials? These are all valuable questions.
2) The figure legends at present are not detailed enough. These need to be fleshed out with all the details, so they can stand alone and be understood.
3) There are a few sentences in which the English/grammar needs improved. For example lines [109] and [240].
Well done on drafting what will be a beautiful, rich, and informative paper.
Author Response
REVIEWER 1
Overall think this paper is fantastically composed. The writing is informative and the figures are beautifully presented with a good level of detail. I have a few points I would like to see addressed to make this paper the best it can be.
- The are a lot of great references to pre-clinical work with cell lines and mice, but I, the reader, want to know more about the current clinical landscape of nanoparticles for radiation therapy. Are they being used clinically at the moment? If not, why not? Are there ongoing clinical trials? These are all valuable questions.
Thank you for your valuable comments. Section 8 was included at the end of the manuscript, to address the missing aspects.
- Potential clinical applications
Several anti-cancer nanosystems have been translated to the clinic. Almost thirty years ago, the liposome/doxorubicin nanosystem was approved by the FDA for cancer treatment, which has demonstrated improved efficacy and safety over standard and single therapies[110]. Since then, at least ten controlled-release nanosystems based on polymeric nanoparticles, micelles, and liposomes have been approved for clinical application, while several more are in clinical phase II and phase III [111]. According to The ClinicalTrials.gov registry, several nanosystems for the delivery of chemotherapeutic drugs for colorectal cancer such as cetuximab loaded into targeted polymeric NP (Phase I), SNB-01 containing the active metabolite of irinotecan (Phase I), folate functionalized nanocarrier for a topoisomerase-1 inhibitor delivery (Phase II) and the Irinotecan liposome formulation (phase II) are under clinical evaluation. Likewise, the CRLX-101 for capecitabine delivery, when combined with conventional therapy and radiation external therapy is under phase I evaluation.
In particular, nanosystems with a dual function of targeted therapy and radiotherapy in colorectal cancer are still under a preclinical stage of research. However, many of the drug delivery radionanosystems previously described in this review have the advantage of being natural or biodegradable vectors that may give the added benefit needed for clinical translation. In other cases, the background of successful radiotherapeutic results has allowed the first application in humans. For example, 177Lu-peptides for targeted radiotherapies of prostate cancer or neuroendocrine tumors [112, 113] served as clinical experience to administer 177Lu2O3-peptide nanoparticles in a patient with colorectal liver metastases[107]. There is no doubt that the preclinical advances in nanosystems shown in this review make it clear that radionanosystems for targeted therapy and radiotherapy of colorectal cancer are on the right path to clinical translation.
2) The figure legends at present are not detailed enough. These need to be fleshed out with all the details, so they can stand alone and be understood.
The legends on figures were detailed, for better understanding.
3) There are a few sentences in which the English/grammar needs improved. For example lines [109] and [240].
Line 109 was changed to:
When metallic nanoparticles are, to reach the colloidal stability is needed. This feature is achieved by minimizing its surface charge, thus reducing the electrostatic interactions.
Line 240 was changed to:
A promising approach is the NBTXR3, proposed by Zhang and cols [51].
Well done on drafting what will be a beautiful, rich, and informative paper.
Thanks for your kind comments.
Reviewer 2 Report
The review is well compiled and in good flow. Following addition can improve overall quality.
- Marketed products
- Case studies where such products used
- Preclinical/clinal data if any
- Patent if any, in tabular form can be better
- Future aspects
Author Response
REVIEWER 2
The review is well compiled and in good flow. Following addition can improve overall quality.
- Marketed products
- Case studies where such products used
- Preclinical/clinal data if any
- Patent if any, in tabular form can be better
- Future aspects
Thank you for your valuable comments. Section 8 was included at the end of the manuscript, in order to address the missing aspects. There are no patents yet reported, which involve nanoparticles for dual treatment of colorectal cancer.
- Potential clinical applications
Several anti-cancer nanosystems have been translated to the clinic. Almost thirty years ago, the liposome/doxorubicin nanosystem was approved by the FDA for cancer treatment, which has demonstrated improved efficacy and safety over standard and single therapies[110]. Since then, at least ten controlled-release nanosystems based on polymeric nanoparticles, micelles, and liposomes have been approved for clinical application, while several more are in clinical phase II and phase III [111]. According to The ClinicalTrials.gov registry, several nanosystems for the delivery of chemotherapeutic drugs for colorectal cancer such as cetuximab loaded into targeted polymeric NP (Phase I), SNB-01 containing the active metabolite of irinotecan (Phase I), folate functionalized nanocarrier for a topoisomerase-1 inhibitor delivery (Phase II) and the Irinotecan liposome formulation (phase II) are under clinical evaluation. Likewise, the CRLX-101 for capecitabine delivery, when combined with conventional therapy and radiation external therapy is under phase I evaluation.
In particular, nanosystems with a dual function of targeted therapy and radiotherapy in colorectal cancer are still under a preclinical stage of research. However, many of the drug delivery radionanosystems previously described in this review have the advantage of being natural or biodegradable vectors that may give the added benefit needed for clinical translation. In other cases, the background of successful radiotherapeutic results has allowed the first application in humans. For example, 177Lu-peptides for targeted radiotherapies of prostate cancer or neuroendocrine tumors [112, 113] served as clinical experience to administer 177Lu2O3-peptide nanoparticles in a patient with colorectal liver metastases[107]. There is no doubt that the preclinical advances in nanosystems shown in this review make it clear that radionanosystems for targeted therapy and radiotherapy of colorectal cancer are on the right path to clinical translation.
Reviewer 3 Report
This is a review paper by Cruz-Nova and coauthors about controlled-release nanosystems with a dual function of targeted therapy and radiotherapy in colorectal cancer. However, several major issue listed below should be addressed properly before it could be considered to be publised in Pharmaceutics.
- The resolution of figure 2 should be much higher. Currently, lots of elements can't seen clearly.
- Line 113, 'Na3C6H5O7' to 'Na3C6H5O7'; line 130, 'I> 30 mVI' to '>I30 mVI'; line 197, 'HfO2' to 'HfO2'; line 251, '10 -1000 nm' to '10-1000 nm'. Please check all.
- Line 271, it would be better to remove '(C2nH4nO2+1)' since the chemical formula will be different if the end group of PEG is different.
- Line 278-279, The mechanism of colloidal stability of PEGylated nanoparticles is attributed to both entropic conformation and enthalpic effects of hydration. Several reviews (Current Opinion in Colloid & Interface Science 44, 94-106; Acta biomaterialia 40 (2016): 6-15) related to this point should be included.
- Section 4.3 DDAB nanosystems is not proper. This is just one example of surfactant (DDAB), however the authors listed it as one category. Meanwhile, DDAB is not a polymer.
- Section 4.4 CRLX101 nanosystem is not general. The authors may use a different term, such as polymer-drug conjugate. CRLX101 would be one example of polymer-drug conjugate.
Author Response
REVIEWER 3
This is a review paper by Cruz-Nova and coauthors about controlled-release nanosystems with a dual function of targeted therapy and radiotherapy in colorectal cancer. However, several major issue listed below should be addressed properly before it could be considered to be publised in Pharmaceutics.
- The resolution of figure 2 should be much higher. Currently, lots of elements can't seen clearly.
Figure 2 was rearranged in order to improve the resolution.
- Line 113, 'Na3C6H5O7' to 'Na3C6H5O7'; line 130, 'I> 30 mVI' to '>I30 mVI'; line 197, 'HfO2' to 'HfO2'; line 251, '10 -1000 nm' to '10-1000 nm'. Please check all.
Done. The changed text is highlighted in yellow. The rest of the paper was also reviewed.
- Line 271, it would be better to remove '(C2nH4nO2+1)' since the chemical formula will be different if the end group of PEG is different.
Thanks for your observation, the text was removed.
- Line 278-279, The mechanism of colloidal stability of PEGylated nanoparticles is attributed to both entropic conformation and enthalpic effects of hydration. Several reviews (Current Opinion in Colloid & Interface Science 44, 94-106; Acta biomaterialia 40 (2016): 6-15) related to this point should be included.
References were reviewed and included
- Section 4.3 DDAB nanosystems is not proper. This is just one example of surfactant (DDAB), however the authors listed it as one category. Meanwhile, DDAB is not a polymer.
The section 4.3 title was corrected to “Poly-ε-caprolactone nanosystems”
- Section 4.4 CRLX101 nanosystem is not general. The authors may use a different term, such as polymer-drug conjugate. CRLX101 would be one example of polymer-drug conjugate.
The section was renamed nanosystems based on polysaccharides.
Round 2
Reviewer 3 Report
I recommend this article for publication.